# Could Combined Action Observation and Motor Imagery Practice, Added to Standard Rehabilitation, Improve Study Upper Limb Functional Recovery in Chronic Stroke Patients? Suggestive Evidence from a Feasability Study

**DOI:** 10.3390/neurosci6040098

**Published:** 2025-10-03

**Authors:** Andrea Peru, Maria Teresa Turano, Barbara Vallotti, Federico Mayer, Costanza Panunzi, Valentina Tosti, Maria Pia Viggiano

**Affiliations:** 1Department of Neuroscience, Psychology, Drug Area and Child Health, University of Firenze, Via di San Salvi 12, 50135 Firenze, Italy; andrea.peru@unifi.it; 2Fondazione Turano Onlus, 00195 Roma, Italy; 3IFCA GIOMI Casa di Cura “Ulivella e Glicini”, Via del Pergolino 4, 50139 Firenze, Italy

**Keywords:** functional recovery, motor recovery, rehabilitation, stroke, upper limb hemiparesis

## Abstract

This study aims to investigate whether a combined action observation–motor imagery practice may enhance the effects of conventional physical rehabilitation in a stroke survivor population. A total of 8 (7 male, 1 female) post-stroke patients with upper limb hemiparesis were enrolled into a single-blinded, randomised, study. Five times per week for three weeks, four patients experienced 60’ conventional physical therapy, while the other 4 experienced 30’ conventional physical therapy and 30’ action observation–motor imagery practice. The Fugl-Meyer Assessment-Upper Extremity and the Wolf Motor Function Test scores from the baseline and post-physiotherapy were used to evaluate upper extremity motor function. Patients who received the AO + MI alongside conventional physical rehabilitation benefitted more than those who received only conventional physical rehabilitation. However, the sample size was very small (only eight participants), which reduces both the statistical power and the ability to generalise the results. Moreover, there was no follow-up; therefore, it is unclear whether the observed improvements lasted over time. Finally, some potentially confounding factors, such as stroke type or lesion site, were not statistically controlled. Notwithstanding these limitations, our findings may serve as a basis for future large-scale, well-controlled studies on AO + MI in stroke rehabilitation.

## 1. Introduction

Stroke represents, globally, the principal cause of long-term disabilities among the worldwide adult population, and upper limb hemiparesis is the most common impairment observed in stroke survivors [1]. Therefore, the exploration of motor rehabilitation techniques represents one of the most relevant clinical challenges to date. For some years now, conventional physical practice has no longer been a gold standard as it was about a decade ago [2]. It is important to note that the traditional rehabilitation approach mainly focusses on passive (a-specific) movement associated with compensatory training of the non-paretic arm [3]. Therefore, the development of training techniques that allow for effective re-activation of the paretic limb has great value. An emerging body of research suggests that two mental practice techniques, action observation (AO) and motor imagery (MI), have potential to improve motor performance in both healthy individuals and brain-damaged patients [4]. While AO refers to the simple observation of an action [5], MI involves internal generation of the visual and kinaesthetic features of movements not accompanied by overt body movements [6].

Thus, MI represents an active state focusing on the kinaesthetic sense of movement during which specific motor action representations are mentally rehearsed without any motor output [7]. Although several studies have suggested that a combined AO + MI practice may be more effective than independent AO or MI, the results from these studies are still controversial. While some studies have provided support for this notion [8,9], other studies have instead found only minimal clinical differences in upper limb motor recovery [10].

Recently, Binks and colleagues [11] re-assessed the issue and argued that in hemiparetic patients, combined AO + MI practice may offer an effective adjunct for the rehabilitation of the upper limb when physical practice is unsuitable. They observed an effect of AO + MI on daily living questionnaires, although no improvement was detected in the quantitative measures of motor function.

Hereafter, we aim to investigate for the first time whether a combined AO + MI practice may foster the effects of conventional physical rehabilitation when assessed using standard outcome quantitative measures of motor function such as the Fugl-Meyer Assessment-Upper Extremity (FMA-UE) [12] and the Wolf Motor Function Test (WMFT) [13].

## 2. Method

### 2.1. Participants

Participants were recruited from a series of brain-damaged patients consecutively admitted to the local neuro-rehabilitation unit (The neuro-rehabilitation unit is part of a private clinic affiliated with the National Health System. Patients discharged from the Stroke Units of the public hospital after the acute post-onset phase are admitted to this clinic to undergo rehabilitation treatments that usually last 1–2 months. Participants in our study were recruited from a series of consecutively admitted patients in the clinic who met the inclusion criteria. All patients underwent the “routine” program (i.e., treatment for 5 days per week for 3 weeks).). Inclusion criteria were patients > 18 years with upper limb hemiparesis because of a single, unilateral stroke that occurred 3–4 months before. Exclusion criteria were an inability to understand instructions and/or a diagnosed neurological condition affecting muscle and joint motion (e.g., spasticity). A total of 8 (7M, 1F) patients (mean age = 68,7; sd = 9.9) met the inclusion criteria. One of them had a right hemisphere lesion, while the remaining seven had a left hemisphere lesion. Five of them presented with ischemic strokes and three with haemorrhagic strokes. None of them had significant mood disorders or cognitive deficits (an MMSE score well above the cut-off). Before starting the experimental investigation, all participants were informed about the general aims of the study and clearly told that participation was not mandatory, and they could withdraw from it at any time without problems. All of them, however, accepted to take part in the study by signing a written consent form.

### 2.2. Motor Function Evaluation

In both the pre- and post-training phases, the patient’s upper limb motor function was assessed by means of the FMA-UE and the WMFT by an examiner who was unaware of the group to which the patients belonged (see below).

The FMA-UE provides an assessment of stroke-related upper limb motor impairments and is often used in rehabilitation research studies [14]. The 33 items of the assessment are scored on a 3-point rating scale: 0 = unable to perform, 1 = partial ability to perform, and 2 = near-normal ability to perform.

In turn, the WMFT [12] quantifies upper extremity (UE) motor ability through timed and functional tasks. It consists of 17 items that should be performed as quickly as possible, truncated at 120 s. Each item is rated on a 6-point scale ranging from 0 (“does not attempt with UE being tested”) to 5 (“does attempt, movement appears to be normal”).

### 2.3. Treatment

An independent research assistant, blinded to the treatment, randomly assigned four patients to the group who experienced conventional physical therapy and AO + MI practice (i.e., the experimental group, in brief) and four to the group who experienced conventional physical therapy only (i.e., the control group, in brief). Non-parametric statistics (Mann–Whitney U) demonstrated that at the baseline, the two groups did not differ from each other in terms of age (*p* = 0.47).

Training was completed five times per week for 3 weeks. In each training session, patients from the control group experienced 60’ conventional physical therapy, while patients from the experimental group experienced 30’ conventional physical therapy and 30’ AO + MI practice.

The AO + MI treatment consisted of three subsequent phases. In the first phase (action observation), patients were asked to observe two 2’ video-clips of object-directed, mono- and bi-manual actions with ecological value (e.g., picking up a glass and drinking). In the second phase (motor imagery), patients had to imagine themselves completing the actions just seen and to verbalise each motor act. For instance, the action of ‘picking up a glass and drinking’ could be divided into 8 motor acts: 1. bringing the upper limb near the glass; 2. opening one’s fingers to grasp the glass; 3. closing one’s fingers to catch the glass; 4. raising the upper limb to bring the glass to his/her mouth; 5. turning the wrist to drink; 6. turning the wrist to return into a vertical position; 7. extending the elbow; 8. lowering the upper limb to put the glass on the table. Finally, in the last phase (motor execution), patients were required to imitate the observed actions to the best of their abilities.

## 3. Results

As reported in Table 1, the rehabilitation treatment improved the motor function of all patients both when assessed with the FMA-UE and WMFT scales.

Baseline differences between groups as well as the change in performance (post-training minus baseline scores) were assessed using the Wilcoxon rank-sum test. The change from the baseline to post-training was calculated and compared between the experimental and control groups using the Wilcoxon signed rank test.

At the baseline, there were no significant differences between the two groups in both FMA-UE (U = 5, *p* = 0.486) and WMFT scores (U = 7.5, *p* = 0.999).

Comparisons between the baseline and post-training assessments using the Wilcoxon signed-rank test revealed improvements in motor function in both groups, as indicated by significant changes in the FMA-UE (W = 0, *p* = 0.014) and WMFT (W = 0, *p* = 0.008) scores. However, the improvement was more pronounced in the experimental group compared to the control group, as clearly illustrated in Figure 1. Specifically, when analysing the change in performance (post-training minus baseline scores), the independent-sample Wilcoxon rank-sum test showed that the treatment induced significant improvements in the experimental group compared to those in the control group only in the WMFT (U = 16, *p* = 0.029). With regard to the FMA-UE, a descriptive difference was noted, but it was not statistically significant (U = 13.5, *p* = 0.146).

## 4. Discussion

The goal of stroke rehabilitation is to help patients recover as much function as possible and maximise their independence and quality of life [15]. In recent years, conventional physical practice has ceased to be considered the gold standard in the rehabilitation of the paretic limb, as it was until about a decade ago [2]. Recently, AO and MI trainings have been proven as promising techniques for upper limb rehabilitation [16,17]. Furthermore, a recent review on the neurophysiological and behavioural evidence of the effects of combined action observation and motor imagery (AO + MI) on motor processes suggests that this combined approach of AO + MI can be more effective in motor learning and rehabilitation settings, relative to the more traditional application of MI or AO alone [4]. However, further evidence is required to substantiate this claim. In this vein, our study aimed to investigate whether a combined action observation–motor imagery (AO + MI) practice could enhance the effects of conventional physical rehabilitation in a stroke survivor population with upper limb hemiparesis. We made a direct comparison of the effectiveness of AO + MI integrated training vs. conventional physical rehabilitation in promoting early recovery of upper limb hemiparesis. For each patient, motor function was evaluated before and at the end of the training by means of two scales widely used in rehabilitation research studies [14].

The results were clear-cut: the rehabilitation treatment improved the motor function of all patients, but the amelioration was much greater for patients who underwent the AO + MI treatment than for those who only experienced conventional physical therapy. These findings align with those of other studies conducted on larger cohorts, which have reported similar outcomes. Conversely, they differ from studies that found no significant effects or only minimal improvements, likely due to methodological discrepancies.

However, some intrinsic limitations of this study suggest caution before drawing any definite conclusion. First, the sample size was very small (only eight participants), which reduces both the statistical power and the ability to generalise the results. Moreover, there was no follow-up, and thus, it is unclear whether the observed improvements lasted over time. Finally, some potentially confounding factors, such as stroke type or lesion site, were not considered. Notwithstanding these limitations, our findings strongly support the integration of AO and MI as a valuable complement into physical therapy in post-stroke hemiparesis rehabilitation at least, for inpatients in the post-acute phase without significant spasticity, cognitive impairment, or mood dysfunction.

One last point deserves consideration: what are the mechanisms involved in the motor recovery fostered by AO + MI training? Our study did not provide any direct evidence on this issue, and the answer can only be speculative. In this vein, involvement of the mirror neuron system seems likely [18]. This system is active during action observation and plays a critical role in understanding the action of others “from the inside”. Thus, we can suppose that patients who undergo AO + MI training develop strategies that allow for the reconstruction of the relative mental representation of the movement observed, resulting in a positive effect on the recovery of the action limb [19]. Moreover, observing meaningful actions (e.g., grasping a glass) and imagining the action may increase motivation and adherence to the rehabilitation program. In this regard, a very recent review emphasises how the integration of different approaches can enhance cognitive, emotional, and behavioural results for individuals with dementia [20].

## 5. Conclusions

Patients admitted to the neuro-rehabilitation unit after the acute post-onset phase were divided into two groups: the first group experienced conventional physical therapy and AO + MI practice, while the second group experienced conventional physical therapy only. Patients who received the AO + MI alongside conventional physical rehabilitation benefitted more than those who received conventional physical rehabilitation alone. However, the sample size was very small, there was no follow-up, and some potentially confounding factors, such as stroke type or lesion site, were not statistically controlled. Notwithstanding these limitations, these findings may serve as a basis for future large-scale, well-controlled studies on AO + MI in stroke rehabilitation.

## Figures and Tables

**Figure 1 neurosci-06-00098-f001:**
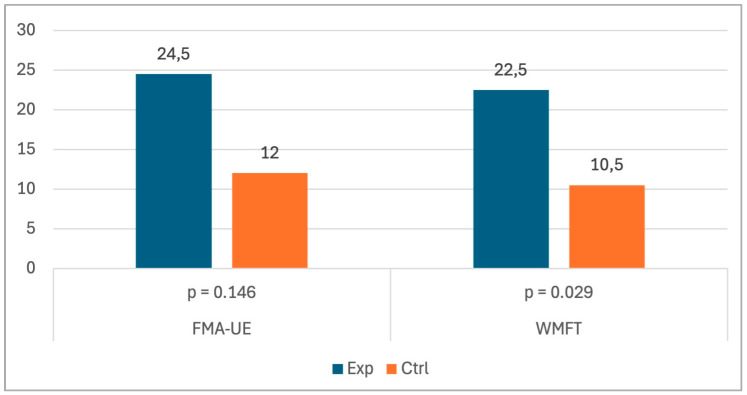
Mean improvement across groups.

**Table 1 neurosci-06-00098-t001:** Patients’ scores at baseline and post-training.

	FMA-UE (Max Score = 66)	*p* = 0.0014	WMFT (Max Score = 85)	*p* = 0.008
Patients	Baseline	Post-Training	Baseline	Post-Training
Exp 1	34	57	34	55
Exp 2	28	47	31	51
Exp 3	31	57	38	62
Exp 4	16	53	33	67
Ctrl 1	44	54	36	49
Ctrl 2	30	56	31	45
Ctrl 3	53	56	49	54
Ctrl 4	19	33	21	29

## Data Availability

The data is no longer available following the collapse of a wing of the department where the paper protocols were stored.

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
