# Peer review of "Could Combined Action Observation and Motor Imagery Practice, Added to Standard Rehabilitation, Improve Study Upper Limb Functional Recovery in Chronic Stroke Patients? Suggestive Evidence from a Feasability Study"

_neurosci, 2025, doi:10.3390/neurosci6040098_

Round 1
Reviewer 1 Report
Comments and Suggestions for Authors
The authors are to be congratulated on this exploring this topic with a straightforward and quite pragmatic methodology, and in a growing area of interest where this type of research can really add value. The paper is easy to read, and provides some clear messages.
The main purpose of this study was, apparently, to determine whether adding Motor Imagery (MI) AND Action Observation (AO) to 'conventional Rehab improves upper limb outcomes for stroke patients. The answer appears to be 'probably yes'. There are however still a lot of uncertainties that make it more difficult to understand how this research can be applied.
There is very little information available to actually define the patient group studied. Knowing that these are stroke survivors with residual upper limb limitations 3-4 months after stroke, we then need to try to interpret the degree of limitation by exploring the limited information around the baseline FMA (UL) and WMFT data presented in table 1. We are told that the patients have been recruited from 'the local neuro-rehabilitation clinic', but we are not informed about the typical (stroke) casemix who attend this clinic, nor even if it is an inpatient or outpatient 'clinic', is it publicly or privately funded? There is no clear information about how the patients were recruited; there is no information about whether these people were consecutive (relevant) presentations, or whether there were a number who were not included for reasons other than the 'exclusion criteria', (e.g. no consent, etc). Better exploration of recruitment process, patient selection/exclusion, time periods for recruitment, patient demographics, etc would be valuable.
Defining the researched population is important. Apart from stating that 'age' was equally distributed between the Intervention and the control groups, we know nothing about a range of other factors that potentially could play a role in outcomes, e.g. spasticity, mood, cognition, inattention, communication, co-morbidities, and more. Without this, it is difficult to interpret who this research is most relevant for.
It is noted that the patients received treatments 5 days per week for 3 weeks. This is not uncommon for inpatient rehab programs, but inpatient programs are not so often available after 3 months post stroke (perhaps more common in privately funded Rehab Units?) While the study appears to demonstrate benefit of MI + AO, it is helpful to know if this benefit was achieved with 'routine' intensity, or was the intensity 'increased' for the purposes of the study. Inpatient vs outpatient delivery also has implications. Many outpatient rehab units would not be able to deliver 5 days/week therapy for 3 weeks.
Data is presented clearly, but minimally. It would be good to see not just 'raw' data (Table 1) but also the data after statistical analysis, as presented in the prose of the Results section, but preferably also in table form. It is more difficult to interpret the data presented in Fig 1, for example, without having the statistical significance presented in the same Figure.
In the Discussion section, it would be good to see more discussion around not only the improvements recorded, and the statistical significance, but also the clinical/functional significance.
The discussion section could also expand more on the limitations of the study. The 'small numbers', as mentioned in the prose, are a definite limitation, but there is minimal exploration about what these limitations are, and how they may impact on any conclusions. There is no real discussion on biases. There is inadequate information to allow any interpretation for generalisability, or applicability.
There are some good references, but some are older (although still may have some relevance). A very recent Systematic Review (Calderone A et al; Biomedicines 2025) may be a good source of some other more recent references that may add value to this paper.
Author Response
The authors are to be congratulated on this exploring this topic with a straightforward and quite pragmatic methodology, and in a growing area of interest where this type of research can really add value. The paper is easy to read, and provides some clear messages.
The main purpose of this study was, apparently, to determine whether adding Motor Imagery (MI) AND Action Observation (AO) to 'conventional Rehab improves upper limb outcomes for stroke patients. The answer appears to be 'probably yes'. There are however still a lot of uncertainties that make it more difficult to understand how this research can be applied.
There is very little information available to actually define the patient group studied. Knowing that these are stroke survivors with residual upper limb limitations 3-4 months after stroke, we then need to try to interpret the degree of limitation by exploring the limited information around the baseline FMA (UL) and WMFT data presented in table 1. We are told that the patients have been recruited from 'the local neuro-rehabilitation clinic', but we are not informed about the typical (stroke) casemix who attend this clinic, nor even if it is an inpatient or outpatient 'clinic', is it publicly or privately funded? There is no clear information about how the patients were recruited; there is no information about whether these people were consecutive (relevant) presentations, or whether there were a number who were not included for reasons other than the 'exclusion criteria', (e.g. no consent, etc). Better exploration of recruitment process, patient selection/exclusion, time periods for recruitment, patient demographics, etc would be valuable.
Defining the researched population is important. Apart from stating that 'age' was equally distributed between the Intervention and the control groups, we know nothing about a range of other factors that potentially could play a role in outcomes, e.g. spasticity, mood, cognition, inattention, communication, co-morbidities, and more. Without this, it is difficult to interpret who this research is most relevant for.
It is noted that the patients received treatments 5 days per week for 3 weeks. This is not uncommon for inpatient rehab programs, but inpatient programs are not so often available after 3 months post stroke (perhaps more common in privately funded Rehab Units?) While the study appears to demonstrate benefit of MI + AO, it is helpful to know if this benefit was achieved with 'routine' intensity, or was the intensity 'increased' for the purposes of the study. Inpatient vs outpatient delivery also has implications. Many outpatient rehab units would not be able to deliver 5 days/week therapy for 3 weeks.
Data is presented clearly, but minimally. It would be good to see not just 'raw' data (Table 1) but also the data after statistical analysis, as presented in the prose of the Results section, but preferably also in table form. It is more difficult to interpret the data presented in Fig 1, for example, without having the statistical significance presented in the same Figure.
In the Discussion section, it would be good to see more discussion around not only the improvements recorded, and the statistical significance, but also the clinical/functional significance.
The discussion section could also expand more on the limitations of the study. The 'small numbers', as mentioned in the prose, are a definite limitation, but there is minimal exploration about what these limitations are, and how they may impact on any conclusions. There is no real discussion on biases. There is inadequate information to allow any interpretation for generalisability, or applicability.
There are some good references, but some are older (although still may have some relevance). A very recent Systematic Review (Calderone A et al; Biomedicines 2025) may be a good source of some other more recent references that may add value to this paper.
Reviewer 1
- We are told that the patients have been recruited from 'the local neuro-rehabilitation clinic', but we are not informed about the typical (stroke) casemix who attend this clinic, nor even if it is an inpatient or outpatient 'clinic', is it publicly or privately funded?
- There is no clear information about how the patients were recruited; there is no information about whether these people were consecutive (relevant) presentations, or whether there were a number who were not included for reasons other than the 'exclusion criteria', (e.g. no consent, etc).
- There is no clear information about how the patients were recruited; there is no information about whether these people were consecutive (relevant) presentations, or whether there were a number who were not included for reasons other than the 'exclusion criteria', (e.g. no consent, etc).
We have reported further information about the neuro-rehabilitation unit, the patients, and the recruitment process in a footnote.
- Apart from stating that 'age' was equally distributed between the Intervention and the control groups, we know nothing about a range of other factors that potentially could play a role in outcomes, e.g. spasticity, mood, cognition, inattention, communication, co-morbidities, and more. Without this, it is difficult to interpret who this research is most relevant for
As requested by the reviewer, we provided additional information on other factors that may play a role in the outcome. In particular, we clearly stated that the presence of spasticity and communication disorders was an exclusion criterion. Furthermore, we reported that none of the patients suffered from mood disorder or cognitive deficits.
- It is noted that the patients received treatments 5 days per week for 3 weeks. This is not uncommon for inpatient rehab programs, but inpatient programs are not so often available after 3 months post stroke (perhaps more common in privately funded Rehab Units?) While the study appears to demonstrate benefit of MI + AO, it is helpful to know if this benefit was achieved with 'routine' intensity or was the intensity 'increased' for the purposes of the study. Inpatient vs outpatient delivery also has implications. Many outpatient rehab units would not be able to deliver 5 days/week therapy for 3 weeks.
The participants were in-patients who underwent the “routine” program (i.e., 5 days per week for 3 weeks).
- It would be good to see not just 'raw' data (Table 1) but also the data after statistical analysis, as presented in the prose of the Results section, but preferably also in table form. It is more difficult to interpret the data presented in Fig 1, for example, without having the statistical significance presented in the same Figure.
Figure 1 has been modified according to the Referee’s suggestion.
- In the Discussion section, it would be good to see more discussion around not only the improvements recorded, and the statistical significance, but also the clinical/functional significance.
The Discussion section has been thoroughly revised.
- The discussion section could also expand more on the limitations of the study. The 'small numbers', as mentioned in the prose, are a definite limitation, but there is minimal exploration about what these limitations are, and how they may impact on any conclusions. There is no real discussion on biases. There is inadequate information to allow any interpretation for generalisability, or applicability.
According to the Referee’s suggestion, the limitations of the study are clearly acknowledged in the Discussion section.
.
8 There are some good references, but some are older (although still may have some relevance). A very recent Systematic Review (Calderone A et al; Biomedicines 2025) may be a good source of some other more recent references that may add value to this paper.
This reference has been added.
Reviewer 2 Report
Comments and Suggestions for Authors
Combined action observation-motor imagery practice may enhance functional rehabilitation in hemiparetic patients: evidence from a preliminary study
General comments
This small clinical trial addresses an interesting addition to what some would consider conventional physical rehabilitation of patients in the chronic phase post stroke. Action observation-motor imagery is not always included in standard rehabilitation programs for patients post stroke. It certainly would be important to document if this strategy was a positive and significant addition to recovery of function of the upper limb in patients post stroke. The manuscript is brief and easy to read.
Concerns.
Maybe the title needs to be a question: Could combined action observation-motor imagery practice added to standard rehabilitation enhance functional recovery of the upper limb in patients chronic post stroke: A small preliminary study
First a hypothesis statement is necessary. This Ho should be measurable. Even in a preliminary study, it should not be a fishing expedition.
In the methods,
The authors should have provided a power analysis which would tell them how many subjects were needed to find a significant difference.
There should be a clear description of the statistical analyses applied throughout. The differences at baseline were tested with the Mann-Whitney Test and then the change pre to post treatment was analyzed with the Welch’s t-test. The Welch’s t tests assumes the variance is not the same in the two groups. However, with such a small number of subjects, non parametric stats like the Mann Whitney or the 2 sample Wilcoxon should have been applied. These assume the variance is not equal and the distribution is not normal. Thus, the tests are based on ranked data. It is really not possible to reach statistical significance with only 4 subjects in each group (.5x.5x.5x.5x=0.0625). It is barely but possible to find significance with 5 in each group (.5x.5x.5x.5x.5=0.03125)
When using non parametric statistics, the data is ranked and reported according to the procedures as the difference in the ranks. If one wanted to create a bar graph, the bar graph would either be the median ( not the mean) or the mean of the ranks.
In the discussion, more elaboration in needed on the limitations of this preliminary or pilot study. What happened to the 9th subject? You need to explain why the patient dropped out or was not included. Why couldn’t you get more patients? Could you add more patients now? As much as desired, it is impossible to generalize the findings in any way. With such a small number, I am not sure that clinicians or researchers would be motivated to repeat the study.
Author Response
Reviewer 2
- Maybe the title needs to be a question: Could combined action observation-motor imagery practice added to standard rehabilitation enhance functional recovery of the upper limb in patients chronic post stroke: A small preliminary study?
- First a hypothesis statement is necessary. This Ho should be measurable. Even in a preliminary study, it should not be a fishing expedition.
The title has been changed. According to the Referee’s suggestion, the title now reads: “Could combined action observation and motor imagery practice, added to standard rehabilitation, improve study upper limb functional recovery in chronic stroke patients? Suggestive evidence from a preliminary study”.
- The authors should have provided a power analysis which would tell them how many subjects were needed to find a significant difference.
Using the Wilcoxon rank-sum test and assuming a medium effect size (d = .5) with an alpha of .05, 67 participants per group are required to achieve a power of .80. It is evident that the present sample is severely underpowered. However, it should be noted that the clinical population under study is extremely rare.
- There should be a clear description of the statistical analyses applied throughout. The differences at baseline were tested with the Mann-Whitney Test and then the change pre to post treatment was analyzed with the Welch’s t-test. The Welch’s t tests assumes the variance is not the same in the two groups. However, with such a small number of subjects, non parametric stats like the Mann Whitney or the 2 sample Wilcoxon should have been applied. These assume the variance is not equal and the distribution is not normal. Thus, the tests are based on ranked data. It is really not possible to reach statistical significance with only 4 subjects in each group (.5x.5x.5x.5x=0.0625). It is barely but possible to find significance with 5 in each group (.5x.5x.5x.5x.5=0.03125).
- When using non parametric statistics, the data is ranked and reported according to the procedures as the difference in the ranks. If one wanted to create a bar graph, the bar graph would either be the median ( not the mean) or the mean of the ranks.
The statistical analyses were revised by applying, as appropriate, the Wilcoxon rank-sum test and the Wilcoxon signed-rank test. After modifying the tests, one of the analyses is no longer statistically significant (although the descriptive data still show a difference consistent with the hypotheses). The discussion was revised accordingly. In addition, Figure 1 was modified to present the median value.
- In the discussion, more elaboration in needed on the limitations of this preliminary or pilot study. What happened to the 9th subject? You need to explain why the patient dropped out or was not included. Why couldn’t you get more patients? Could you add more patients now? As much as desired, it is impossible to generalize the findings in any way. With such a small number, I am not sure that clinicians or researchers would be motivated to repeat the study.
According to the Referee’s suggestion, the limitations of the study are clearly acknowledged in the Discussion section.
Reviewer 3 Report
Comments and Suggestions for Authors
The study addresses an important and relevant question, clearly describing the Action Observation and Motor Imagery (AO+MI) intervention and sharing encouraging preliminary results. However, a few limitations should be considered when interpreting the findings. The sample size was very small (only eight participants), which reduces both statistical strength and the ability to generalize the results. Because there was no follow-up, it’s unclear whether the observed improvements lasted over time. Details on randomization and allocation concealment are described only briefly, raising some uncertainty about potential selection bias. There were also baseline differences between groups and other potentially important factors, such as stroke type or lesion site, were not statistically controlled.
I recommend acceptance with minor revisions, on the condition that these limitations are integrated into both the abstract and discussion to make their implications clear. Expanding the description of the randomization process and examining potential confounding factors in more depth would enhance the clarity of the paper. These refinements would position the manuscript to serve as a basis for future large-scale, well-controlled studies on AO+MI in stroke rehabilitation.
Author Response
Reviewer 3
- Expanding the description of the randomization process and examining potential confounding factors in more depth would enhance the clarity of the paper.
We are grateful to Reviewer 3 for giving us the opportunity to clarify this point. We acknowledged the limitations of the study in both the abstract and discussion and clearly stated that our findings are far from conclusive.
Round 2
Reviewer 1 Report
Comments and Suggestions for Authors
The authors have improved the paper with their revisions. The paper contains information and conclusions that add worthwhile data and outcomes for the topic under review, and this is appreciated.
It would be nice to see that the 'patirent casemix' in this study is (re-) defined in the Discussion, e.g. "...our findings strongly support.....in post-stroke hemiparesis rehabilitation for inpatients in the post-acute phase without significant spasticity, cognitive impairment, or mood dysfunction...", or other words as determined by the authors. This will assist readers to know who to apply the therapy to, and also helps (other) researchers to know who/where to focus their potential interventions. It is also a more accurate description of the outcome /conclusion of this study.
It would be easier for the reader to interpret the data in Table 1, and Figure 1 if the p-values (at least) were included in the actual Table / Figure, rather than just in the prose.
Once again, the authors are to be congratulated on delivering this research.
Author Response
It would be nice to see that the 'patirent casemix' in this study is (re-) defined in the Discussion, e.g. "...our findings strongly support.....in post-stroke hemiparesis rehabilitation for inpatients in the post-acute phase without significant spasticity, cognitive impairment, or mood dysfunction...", or other words as determined by the authors. This will assist readers to know who to apply the therapy to, and also helps (other) researchers to know who/where to focus their potential interventions. It is also a more accurate description of the outcome /conclusion of this study
Done
- It would be easier for the reader to interpret the data in Table 1, and Figure 1 if the p-values (at least) were included in the actual Table / Figure, rather than just in the prose
According to the Reviewer's suggestion, p-values have been included in Table 1 and Figure 1
Reviewer 2 Report
Comments and Suggestions for Authors
The authors have improved the manuscript by addressing the recommendations of the reviewer. I hope there is a journal that would be interested in their preliminary study. However, I believe that this preliminary study of a very interesting question cannot be published with such a small sample of subjects. The journal is a respectable journal and should be embarassed to publish a study with such a small number of subjects. The power analysis supports my comments. Unless they can recruit up to 7 subjects in each group, this pilot study should not be published.
Author Response
- The authors have improved the manuscript by addressing the recommendations of the reviewer. I hope there is a journal that would be interested in their preliminary study. However, I believe that this preliminary study of a very interesting question cannot be published with such a small sample of subjects. The journal is a respectable journal and should be embarassed to publish a study with such a small number of subjects. The power analysis supports my comments. Unless they can recruit up to 7 subjects in each group, this pilot study should not be published.
This decision is up to the Editor
Reviewer 3 Report
Comments and Suggestions for Authors
I thank the authors for their response. The manuscript is suitable for acceptance in its current form.
Author Response
- I thank the authors for their response. The manuscript is suitable for acceptance in its current form
We are grateful to this Reviewer for his/her appreciation of our work in preparing the revised version of our manuscript.